# Improving Millennial Employee Well-Being and Task Performance in the Hospitality Industry: The Interactive Effects of HRM and Responsible Leadership

**Jie He [1]**, **Alastair M. Morrison [2]** and **Hao Zhang [3],***

[1]  Hunan Institute for Innovation and Development, School of Business, Hunan University of Science and Technology, Xiangtan 411201, China
[2]  International College, National Kaohsiung University of Hospitality and Tourism, Gaoxiong 81271, Taiwan
[3]  School of Tourism and Hospitality Management, Hubei University of Economics, Wuhan 430205, China
*   Correspondence: hzhang@hbue.edu.cn

**Abstract:** The purpose of this study is to determine the influence of organizations on millennial (born 1981–2000) employee well-being and task performance in the hospitality industry. Millennial employees are becoming the main workforce in hotels and their well-being is drawing greater attention in research and practice. Prior studies suggest that human resource management (HRM) bridges the organization and employees and has a significant influence on employee well-being in the hospitality industry. Additionally, the supervisor's responsible leadership is a key factor influencing employees in a changing workforce. However, how do HRM and responsible leadership contribute to millennial employee well-being? Can employee well-being make a difference in the hospitality industry? Those questions remain unanswered. To address the two questions, this research aims to examine the interaction effects of HRM and responsible leadership on millennial employee well-being and the mediating effect of well-being between the HRM and responsible leadership on employee performance. The research hypotheses were tested with multiple linear regression analysis based on a survey of 243 millennial employees in the Chinese hospitality industry. The results indicated that HRM and responsible leadership have positive impacts on millennial employee well-being, which could improve task performance in the hospitality sector. In addition, responsible leadership strengthens the positive effect of HRM on employee well-being and task performance. In addition, the interaction effects of HRM and responsible leadership on performance are mediated by employee well-being. It is of practical value for hotels to know that millennial employee well-being can be fostered through improving HRM and responsible leadership.

**Keywords:** responsible leadership; human resource management (HRM); millennial employee well-being; task performance; China

---

## 1. Introduction

High stress, work–life conflict, employee burnout and high levels of turnover represent great challenges to human resource management (HRM) in hospitality [1–5]. It is well known that the hospitality industry is a service-based and labor-intensive sector. Evidence shows that employee well-being, characterized as the assessment of employee experiences based on their perceptions of life and work, is positively related with job satisfaction, organizational commitment, and individual task performance [6–9], and negatively related with turnover intention and burnout [10–12]. The call for fostering greater employee well-being in the hospitality industry is attracting more attention from

researches and practitioners, especially as a prime HRM goal in several respectable hotel groups including Four Seasons, Hyatt, and Banyan Tree.

Importantly, the pursuit of well-being is increasingly being emphasized by millennial (born 1981–2000) employees in the hospitality industry [13–15]. Millennial (also often called Generation Y) employees differ in work values, attitudes, expectations, preferences and concerns from previous generations [16,17]. Generally speaking, millennial employees in hospitality have a greater desire for recognition, respect, continuous development, work involvement, supportive management, fairness, tolerance, and concern for individual welfare, work–life balance, and well-being [13,18–20]. However, employment in the hospitality industry in China is characterized by long work hours, low status, low pay, work–family conflicts, and high employee turnover, despite the rapid development of the sector [8,17]. Therefore, it is a greater challenge to manage millennial hospitality employees, especially with respect to HRM practices and supervisory leadership models [21].

There are conflicts between millennial employee concerns for well-being and negative employment in the hospitality industry [13,20]. Therefore, several recent trends including high levels of human resource outflows and low career commitment have created unique challenges for the hospitality industry in achieving a positive impact on well-being. It is crucial for the sustainable development of the sector to enhance millennial employee well-being to attract, retain and develop younger staff [22]. Can organizations meaningfully and significantly contribute to employee well-being in the hospitality sector? Indeed, it is well acknowledged that organizations are responsible for improving employee well-being ("being happy, healthy and prosperous," Merriam-Webster), especially via the contributions of HRM and supervisory leadership.

Researchers suggest that HRM and supervisory leadership are key antecedents of employee well-being in the workplace [23–26]. HRM refers to a bundle of interconnected organizational practices that influence employee work motivation, competence, and opportunity [23]. HRM, as the bridge between the hotel and employees, is a critical source of employee well-being in the workplace through system and process management [27–30]. Supervisory leadership has significant effects on employee well-being through relationships between individuals [31,32]. More specifically, responsible supervisory leadership refers to leaders treating their employees as stakeholders to improve bonding between organizations and employees [33–35]. Responsible leadership is required in hotels due to the increasing complexity of stakeholder demands [36], sustainable value creation [37], and greater consciousness of internal and external stakeholders [38,39]. Therefore, responsible supervisory leaders who sincerely care about the conditions for employees and who regard employees as important stakeholders tend to enhance employee well-being. In addition, research and practice have demonstrated that HRM and leadership are not separate but interact to influence employee outcomes [24,40].

Despite the research emphasizing the effects of HRM and responsible supervisory leadership on employee well-being from differing perspectives, two important questions remain unclear. First, the question as to how the interaction of HRM and responsible leadership influences employee well-being and, consequently, task performance lacks adequate research evidence. Second, millennial employee management is attracting a great deal of attention. However, the interaction effects of HRM and responsible leadership on millennial employee well-being remains under-researched.

The main purpose of this research was to explore the effects of HRM and responsible leadership on millennial employee well-being and, consequently, their resulting task performance in the hospitality industry. The research objectives were to extend the research in two directions. First, the intention was to shed more light on millennial employee management in the hospitality industry by linking organizational factors (HRM and responsible leadership) and task performance, and to explain why the hospitality industry has the responsibility to care about millennial employee well-being. Second, the interaction effects of HRM and responsible leadership on employee well-being and task performance were to be explored.

## 2. Literature Review and Hypotheses

In line with the research purpose and objectives, the literature review included two parts: millennial employees in hospitality and the relationships among key variables. First, millennial employees in hospitality and the principal relationships among HRM, responsible leadership, and employee well-being are discussed based upon the existing literature. Next, the research on the interplay of HRM, responsible leadership, and employee task performance is reviewed. Third, the mediating role of employee well-being between its antecedents and task performance is analyzed.

### 2.1. Millennial Employee Well-Being in Hospitality

Millennials have drawn a great deal of attention in business practice and in the research literature since the publication of the book, "Millennials Rising: The Next Great Generation", by Strauss and Howe in 2000. Millennials represent individuals born between 1981 and 2000 [17,21,41,42]. A generation means an identifiable group that shares similar social and historical experiences, significant life events, and critical developmental stages [14,16].

According to the related research, millennial employees share similar social and historical contexts, are more ambitious and concerned about career development, desire empowerment and autonomy at work, highlight quality of life, and value work–life balance and overall well-being [18,19,38]. Chen and Choi (2008) found that millennial employees in hospitality emphasize comfort and security, and professional growth work values [43]. In addition, Gursoy et al. (2013) and Brown et al. (2015) proposed that millennials in hospitality prefer work–life balance and immediate recognition and tend to challenge authority. Also, millennials are concerned about learning and development opportunities, and value flexibility in the workplace [44], a commitment relationship between the hotel and individual [8], supportive leadership, and meaningful work–life balance [21].

Millennials are becoming the main workforce in the hospitality sector [8,17,38,40]. However, high stress, work–life conflicts, and career development limitations in hospitality are creating greater HRM challenges [1,2,45,46]. Therefore, improving millennial employee well-being is a critical responsibility for developing and retaining staff who demonstrate positive behaviors at work and support sustainable hospitality development [47].

Human beings have been pursuing well-being for centuries. With the development of positive psychology, employee well-being is becoming a central topic in hospitality HRM research. Employee well-being refers to the overall assessment of employee lives based on their subjective perceptions of life [48,49].

With rapid economic growth and an expanding workforce, workplace well-being is increasing in importance in China because working has become a vital part of most people's lives. Workplace well-being is representing a more critical element of overall well-being including work-related positive affect and psychological well-being [13]., which refers to the quality of employees' physical (e.g., headaches and muscular discomfort) and mental experiences (e.g., fatigue, satisfaction, attachment, self-respect, and depression) at work [50,51]. Evidence shows that millennial hospitality employees are more concerned about well-being in the workplace [18,52].

Taking into account cultural differences between the West and East, Zheng et al. (2015) proposed that employee well-being involves the subjective satisfaction of psychological needs, life and work, or more precisely, psychological well-being, life well-being, and workplace well-being [30]. This research defined millennial employee well-being as being composed of life, work, and psychological well-being following Zheng et al. (2015).

### 2.2. HRM and Employee Well-Being

Employee well-being refers to the perceived quality of individual experiences, including quality of life, workplace, and psychological feelings—all of which can be impacted by HRM [30]. HRM refers to an assortment of practices that impact employee work competence including recruitment and selection,

training and development, motivation including performance-based compensation, performance-based appraisal, job security and promotion from within, and opportunities including teamwork, employee advice, and participation in decision making [53,54]. HRM is a vital organizational factor influencing employee perceived well-being [55–57].

Considerable evidence indicates that HRM systems have significant effects on employee well-being through long mutually reciprocal relationships between organizations and individuals [53], including high-performance work systems [58], high-involvement systems [59], and development HRM [25]. However, employee well-being as a by-product in these research studies has not received enough attention, and it is worthwhile to "give greater priority to the impact of HR practices designed to improve well-being on both well-being and performance" (Guest, 2017, p. 26) [17].

Hospitality HRM should not only focus on task performance but also show concern for employee well-being to meet the needs of millennial employees [60–62] (Stewart et al., 2017). However, how HRM benefits both individuals and organizations is still in search of more answers [17]. There are three reasons for the HRM impacts on employee well-being. First, HRM can satisfy employees' psychological and physical needs. HRM approaches including recruitment and selection, training and development, appraisals and incentives based on performance, employee ownership, and job security can improve employee work capabilities, motivation and opportunities to facilitate psychological need satisfaction, leading to employee perceived well-being [63,64].

Second, HRM may improve the climate for employee well-being. Veld and Alfes (2017) found that HRM builds a supportive, committed and caring climate for employees, improving employee well-being [29]. HRM practices such as flexible job design and participation promote supportive and trusting environments for individuals [17], leading to employee well-being [6]. Employee involvement practices can facilitate employee well-being through meaningful work environments and create win–win situations for employers and employees [65].

Third, HRM can improve the psychological connections between organizations and employees. Cooper et al., 2014, Kim (2019) explain that HRM has positive effects on employee workplace happiness and mental health leading to enhanced job engagement in hotels [28,66]. In addition, HRM systems can enhance perceived organizational justice leading to employee well-being [58], or employee security in the organization [61]. With the increasing need for the involvement and development of millennial employees in hospitality, HRM initiatives including training, development, and performance-based appraisal can improve employee well-being in the workplace. Therefore, it is proposed that:

**Hypothesis 1 (H1).** *HRM is positively related to employee well-being.*

*2.3. Responsible Leadership and Employee Well-Being*

Responsible leadership is defined as leadership that emphasizes the firm's sustainable development and embraces social responsibility [67]. It is proposed that responsible leadership considers employees to be important stakeholders and cares about employee benefits, maintains team psychological safety, shares knowledge with stakeholders, and pursues the sustainable development of the organization [68,69]. Responsible leadership balances the benefits of stakeholders, including employees, customers, governance, and society [70].

Evidence shows that responsible leadership positively impacts employee well-being. Responsible leadership provides for day-to-day communications with employees including supporting, coaching, and encouragement, leading to mutually beneficial relationships between leaders and members that impact employee well-being [12,71]. Responsible leadership may enhance employee well-being for two reasons. First, the principle of "doing good" and caring as a priority ethic may increase employee psychological safety and positive individual experiences leading to enhanced well-being in the workplace [31,72]. The line supervisor who leads with high psychological capital facilitates employees' positive experiences including happiness, hope and self-respect, and improves their well-being at work [73,74]. Employees who perceive supportive, respectful, honest, and trustworthy leadership tend

to experience more meaning and self-efficacy and be more engaged, leading to well-being due to less stress and fatigue at work [75].

Responsible leadership promotes employee well-being through positive relationships between leaders and members. Responsible leaders are genuinely concerned about employees and not only care about task performance but also consider employee satisfaction and development to promote employee well-being [1,63]. Supportive and moral leadership can enhance employee well-being through building reciprocal relationships between leaders and subordinates [36,76], and providing support and trust for employees [77,78]. Kara et al. (2013) and Joseph (2019) propose that inspirational motivation, intellectual stimulation, and individualized consideration from the leader fosters positive employee experiences of work life, commitment, and well-being in hospitality [32,79]. Millennial employees have a desire for greater support and respect from supervisors, calling for responsible leadership in the contemporary hospitality business [21,70]. Therefore, it is reasonable to suggest that responsible leadership can improve employee well-being and, thus, the second hypothesis is:

**Hypothesis 2 (H2).** *Responsible leadership is positively associated with employee well-being.*

## 2.4. The Interaction Effects of HRM and Responsible Leadership

There are interaction effects between HRM and responsible leadership on employee experiences in organizations [24,80]. The interaction effects represent a variable that influences the direction and/or strength of the relationship between the independent variable and a dependent variable. This research proposes that responsible leadership affects the strength of the relationship between HRM and employee well-being.

First, it is acknowledged that HRM and leadership are among the most influential organizational factors for employee experiences at workplaces [5,23,25]. HRM is a critical antecedent of employee experience and psychological need satisfaction in the hospitality sector that affects employee work competence, motivation, and opportunities [81,82]. For example, HRM systems emphasizing employee development within organizations and providing organizational support can improve employee well-being [83].

In addition, responsible leadership makes for differences in employee motivation, attitudes, and experiences, including psychological empowerment, trust, commitment and identity in the organization [13,84]. Employee well-being not only refers to physical well-being but also incorporates mental and social well-being in hospitality organizations [8]. Responsible leadership contributes to the latter two components of enhanced well-being, through for example harmonious relationships between superiors and subordinates and reduced mental stress. Based on the conservation of resource theory, responsible leadership characterized as supportive, trustworthy and caring, beneficially affects employee well-being through interpersonal relationships, and positive psychological experiences [77–79].

Therefore, HRM and responsible leadership may function in different ways but together send supportive and caring signals to employees and build reciprocal relationships between organizations and individuals [25]. Leroy et al. (2018) explained that supervisory leadership impacts employee experiences through informal individual relationships including caring, encouraging and offering advice, and HRM influences employees through formal management systems and processes including training and development, promotion, and performance appraisal [24]. Moreover, responsible leadership has a day-to-day focus on caring, supporting, and being trustworthy for employees; HRM encompasses development and commitment practices and systems that are complementary organizational factors improving employee psychological experiences and fostering good relationships in workplaces [25].

In addition, HRM as an organizational system is associated with employee psychological experiences and satisfies basic individual motivations including caring, developing, and supporting. Responsible leadership and HRM have the common goal of developing employee self-respect, happiness, meaningfulness, commitment, and trust in workplaces [79]. Responsible leadership is a

complementary force to HRM and can strengthen its effects in HR development and motivation. With millennial employees' emphasis on the development of harmonious relationships with leaders and organizations, it is worthwhile to extend the two lines of research for hospitality [17]. Therefore, it is proposed that responsible leadership and HRM may have interaction effects on employee well-being in the third hypothesis:

**Hypothesis 3 (H3).** *The interaction effects of HRM and responsible leadership are positively associated with employee well-being.*

### 2.5. The Mediation Effects of Employee Well-Being

The relationship between employee well-being and task performance has attracted attention in HRM research [85,86]. Task performance refers to efficiency and effectiveness in the task [87]. Evidence shows that employees experiencing high levels of well-being including psychological safety, happiness, trust, and work–life balance are inclined to have better task performance [85,88]. Based on the perspectives of job demands and resources and conservation of resource theory, HRM and responsible leadership represent the job resources that improve employee work experiences including commitment, happiness, and meaningfulness leading to better task performance (Bakker and Demerouti, 2018). Millennial employees in hospitality value quality of life and well-being [17]. A high level of well-being includes resource conservation at work, leading to more cognitive and affective investment that facilitates individual task performance [28,30].

The interaction effects of responsible leadership and HRM on employee task performance are mediated by employee well-being. There is an indirect relationship between the interaction effects of leadership and HRM and employee task performance in the workplace [9]. Current research based on competence, motivation and opportunities explains the reasons for HRM and leadership facilitating employee task performance through commitment, caring, and support [23]. How to motivate employees to engage in service delivery is crucial in the hospitality sector. Employee well-being representing a holistic experience in hospitality represents a crucial way to improve task performance [28], especially for millennial employees in terms of committed, caring, and supportive relationships, and well-being [17]. Employees obtaining positive resources from HRM and responsive leadership have enhanced psychological resources and safety that protect against resource loss and enable recovery from resource loss in service delivery so as to improve task performance [36]. Kim et al. (2019) proposed that HRM promotes employee engagement in hospitality work environments through employee well-being including happiness and mental health [28]. Therefore, it is believed that HRM and responsible leadership create more positive employee experiences and favorable attitudes, which in turn improve task performance [25,57]. Based on the conservation of resource theory, it is proposed that millennial employees perceive responsible leadership and HRM as improving task performance through well-being, as the fourth hypothesis:

**Hypothesis 4 (H4).** *Employee well-being mediates the relationship between the interaction effect of HRM and responsible leadership and task performance.*

The conceptual model for this research is shown in Figure 1.

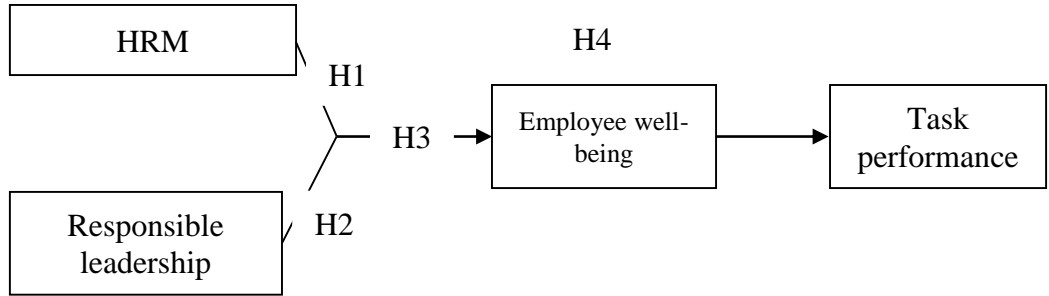

**Figure 1.** Hypothesized conceptual model.

## 3. Methodology

### 3.1. Sample and Procedures

The convenience sampling method was applied for four- and five-star hotels in selected cities of China (Kong et al., 2016) [19]. Questionnaires were sent to 300 employees (born between 1980 and 2000) in hotel companies including InterContinental, Sheraton, Banyan Tree, and Renaissance in Shenzhen, Hangzhou, Shanghai, Beijing, Wuhan and Changsha from November 2017 to January 2018. WeChat, email, and QQ were used in collecting the data. There are long-lasting, cooperative relationships between universities and well-known, high-star level chain hotels in China including Hilton, InterContinental, Hyatt, and Marriott. Additionally, most hospitality management graduates work for these hotel groups. With the assistance of university alumni, questionnaires were sent to millennial employees of these hotels.

A total of 260 responses were received. Of these, 243 valid questionnaires were retained after excluding 17 forms missing key variables or with obvious tendencies in answers (e.g., the same answers for more than eight consecutive questions). Hotel employees from China were chosen for three reasons. First, the Chinese tourism industry is rapidly developing and making a significant contribution to better living standards and economic transformation. Second, service quality and employee behavior are important in Chinese tourism and hospitality. Third, the tourism industry faces significant challenges in social responsibility and sustainable development in China.

The respondents included 30.5% males (75) and 69.5% females (169). On average, they were 24.8 years old, had a 2.3-year tenure, and 228 (93.9%) had college degrees or above. There were 147 front-line employees, accounting for 60.5% of the total sample.

### 3.2. Measures

Five-point Likert scales were used to measure responsible leadership, HRM, employee well-being, and task performance, ranging from "strongly disagree" (1) to "strongly agree" (5). The scales used for these four variables are described in the following:

HRM. HRM was measured with a scale adapted from Dumont et al. (2017) [89]. The items included, "My company provides employees with training to develop employees' knowledge and skills" and "My company provides opportunities for upward mobility." The Cronbach's α was 0.93.

Responsible leadership. The scale for responsible leadership from Voegtlin (2011) was applied [35]. Representative items were, "Our leader is aware of and considers the consequences of one's actions for all stakeholders"; "Our leader tries to achieve a consensus among the participants by weighing the arguments and balancing the interests of the stakeholders." The scale showed good reliability with a Cronbach's α of 0.79.

Employee well-being. The measure of employee well-being including life well-being, work well-being, and psychological well-being was from the Zheng et al. (2015) scale [30]. The items included, "I am close to my dream in most aspects of my life", "I find real enjoyment in my work"; and "I generally feel good about myself, and I am confident." The Cronbach's α was 0.93.

Task performance. The measures for individual task performance focused on task quality, efficiency, and quantity with a 3-item scale adapted from Farh et al. (2007) [87]. The items were, "High quality, low errors, and high accuracy in main job responsibilities", "High efficiency, fast execution, and high quantity in main responsibilities", and "Achieve high goals and in key job responsibilities." The Cronbach's α was 0.78.

Control variables. The researchers controlled for the demographic factors (age, gender, education level, position and tenure, and company ownership) related to employee task performance (Liu et al., 2016).

## 4. Results

### 4.1. Confirmatory Factor Analysis (CFA)

CFA was applied to test the discriminant validity of the variables. The four-factor model showed the acceptable values ($\chi^2/\mathrm{df} = 2.69 < 3$; normed fit index (NFI) = 0.95; non-normed fit index (NNFI) = 0.97; comparative fit index (CFI) = 0.97; incremental fit index (IFI) = 0.97; root mean square error of approximation (RMSEA) = 0.084 < 1). The results indicated that there was significant discrimination among the variables, and the CMV (Common Method Variance) was acceptable. The CFA results show in Table 1.

**Table 1.** Results of Confirmatory Factor Analysis.

| Model. | $\chi^2$ | df | $\chi^2$/df | RMSEA | NFI | NNFI | CFI | IFI |
|---|---|---|---|---|---|---|---|---|
| One-factor | 2122.35 | 405 | 5.24 | 0.132 | 0.93 | 0.94 | 0.94 | 0.94 |
| Two-factor | 1339.70 | 404 | 3.32 | 0.098 | 0.94 | 0.96 | 0.96 | 0.94 |
| Three-factor | 1311.50 | 402 | 3.26 | 0.097 | 0.94 | 0.96 | 0.96 | 0.96 |
| Four-factor | 1075.45 | 399 | 2.69 | 0.084 | 0.95 | 0.97 | 0.97 | 0.97 |

Note. n = 243. RMSEA = root mean square error of approximation, NFI = normed fit index, NNFI = non-normed fit index; CFI = comparative fit index; IFI = incremental fit index. One-factor: HRM (human resource management) + RL (responsible leadership) + WB + P; Two-factor: HRM + RL, WB + P; Three-factor: HRM + RL, WB, P; Four-factor: HRM, RL, WB, P.

In addition, the factor loadings (λ), AVE (average variance extracted) values, and CR (composite reliability) were calculated to examine construct validity. The factor loadings (λ) > 0.5, AVEs > 0.45, and CR > 0.7 showed acceptable validity. The AVE and CR results show in Table 2.

**Table 2.** Results of the average variance extracted (AVE) and composite reliability (CR).

| Variables | Items | λ | CR | Variables | Items | λ | CR |
|---|---|---|---|---|---|---|---|
| Employee well-being | W1 | 0.71 | 0.93 | HRM | HRM1 | 0.79 | 0.94 |
| | W2 | 0.72 | | | HRM2 | 0.82 | |
| | W3 | 0.51 | | | HRM3 | 0.88 | |
| | W4 | 0.76 | | | HRM4 | 0.87 | |
| | W5 | 0.63 | | | HRM5 | 0.85 | |
| | W6 | 0.72 | | | HRM6 | 0.87 | |
| | W7 | 0.79 | | | AVE | 0.72 | |
| | W8 | 0.79 | | Responsible leadership | RL1 | 0.71 | 0.80 |
| | W9 | 0.75 | | | RL2 | 0.57 | |
| | W10 | 0.57 | | | RL3 | 0.67 | |
| | W11 | 0.50 | | | RL4 | 0.64 | |
| | W12 | 0.55 | | | RL5 | 0.74 | |
| | W13 | 0.68 | | | AVE | 0.45 | |
| | W15 | 0.68 | | Task performance | P1 | 0.77 | 0.78 |
| | W16 | 0.69 | | | P2 | 0.68 | |
| | W17 | 0.66 | | | P3 | 0.77 | |
| | AVE | 0.47 | | | AVE | 0.55 | |

Note. W: well-being; HRM: human resource management; RL: responsible leadership; P: performance

### 4.2. Descriptive Statistics

The means, standard deviations, correlations, and reliability statistics for the key variables are presented in Table 3. The variables all possessed acceptable correlations.

**Table 3.** Descriptive statistics and correlations for key variables.

| Variables | Mean | SD | 1 | 2 | 3 | 4 |
|---|---|---|---|---|---|---|
| HRM | 3.46 | 0.84 | 1 | | | |
| Responsible leadership | 3.88 | 0.54 | 0.62 ** | 1 | | |
| Well-being | 3.79 | 0.51 | 0.66 ** | 0.69 ** | 1 | |
| performance | 3.76 | 0.55 | 0.66 ** | 0.60 ** | 0.75 ** | 1 |

Note. Reliability coefficients are shown in parentheses on the diagonal. * $p < 0.05$, ** $p < 0.001$, *** $p < 0.001$, two tailed.

### 4.3. Hypothesis Testing

#### 4.3.1. Mediation Analyses

Lisrel 8.7 was applied to test the mediation effects of employee well-being. The mediation model showed an acceptable fit ($\chi^2 = 1079.96$, df = 401, $\chi^2/df = 2.69 < 3$; NFI = 0.95; NNFI = 0.97; CFI = 0.97; IFI = 0.97; RMSEA = 0.084 < 1). The path estimates showed that the employee well-being played a significant role between HRM and task performance, and responsible leadership and task performance, respectively. The mediation model shows in Figure 2.

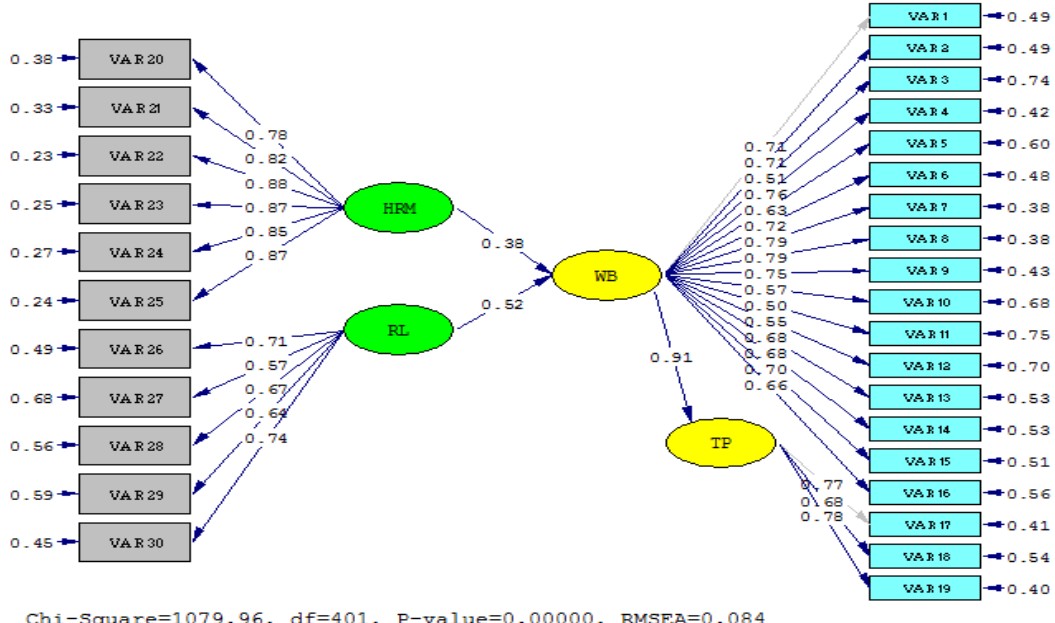

**Figure 2.** Mediation model.

#### 4.3.2. Mediated Moderation Effects

The mediated moderation model was examined following Liu, Zhang and Wang (2012) [90]. First, the relationship between HRM and well-being was examined. Second, the moderating effects of responsible leadership on the relationship between HRM and employee well-being was tested. Third, the mediating effects of employee well-being between the interaction effects and employee task performance were examined. This provides explanations on how responsible leadership changes the relationship between HRM and employee well-being, and the relationship between well-being and task performance.

During step 1, examining the relationship between HRM and employee well-being, the results showed that HRM was positively related to employee well-being (M2: $\beta = 0.29$, $p < 0.001$). Also, there was a positive relationship between responsible leadership and employee well-being. These results supported H1 and H2.

During step 2, the moderating effects of responsible leadership on the relationship between HRM and employee well-being were tested. H3 suggested that responsible leadership strengthened the effect of HRM on employee well-being. The results indicated that the interaction effects of responsible leadership and HRM were positively related with employee well-being (M3: $\beta = 0.13$, $p < 0.01$), supporting H3.

The mediated moderation model testing procedures recommended by Liu et al. (2012) were followed to test Hypotheses 4. First, employee well-being and task performance were regressed after demographic variables were controlled (M5: $\beta = 0.67$, $p < 0.001$). Second, controlling employee well-being, the relationship between HRM, responsible leadership, and task performance was tested. As shown in Model 7, the effects of HRM, responsible leadership, and the interaction effects of the two on task performance were significant (M7: $\beta = 0.37$, $p < 0.001$; $\beta = 0.27$, $p < 0.001$; $\beta = 0.11$, $p < 0.01$). After controlling for employee well-being in Model 7, the effect of HRM, responsible leadership, and the interaction effects on employee task performance were weaker, and the effects of responsible leadership and interaction effects were not significant. Therefore, there is a significant and indirect relationship between the antecedents and the outcomes, and the effect of HRM and responsible leadership on performance was partially mediated by task performance. In addition, the bootstrap test confirmed that the mediation effect of employee well-being was significant (M6: 0.39, 0.74, $p < 0.05$). H4 was supported. The mediated moderation effects are presented in Table 4.

The interaction effects of HRM and responsible leadership on employee well-being and task performance are depicted in Figures 3 and 4:

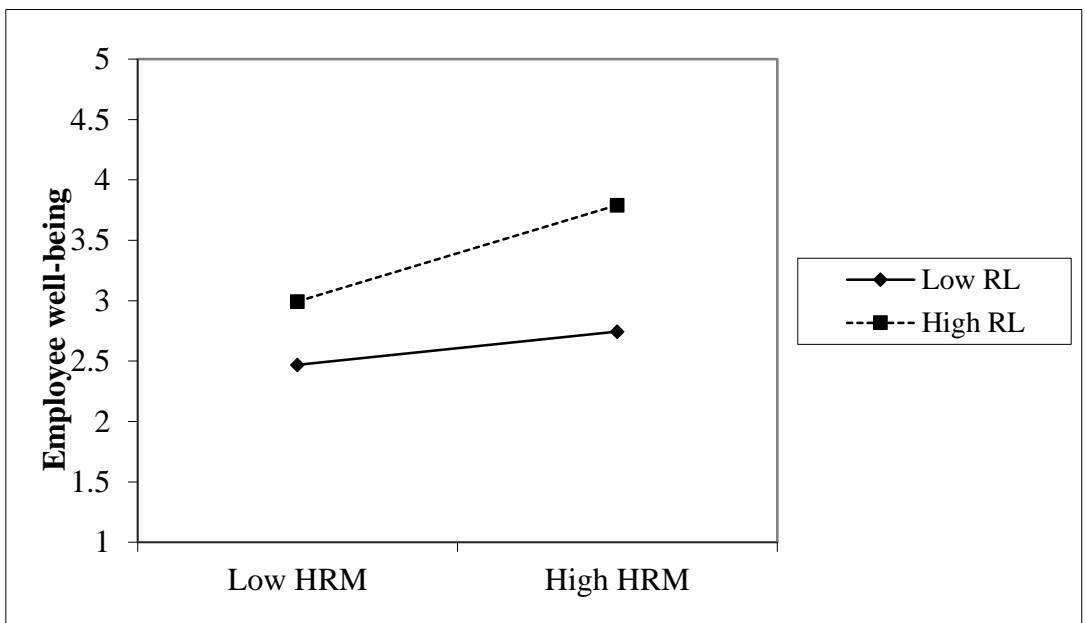

**Figure 3.** The interaction effects of HRM and responsible leadership on employee well-being. HRM: Human resource management; RL: Responsible leadership.

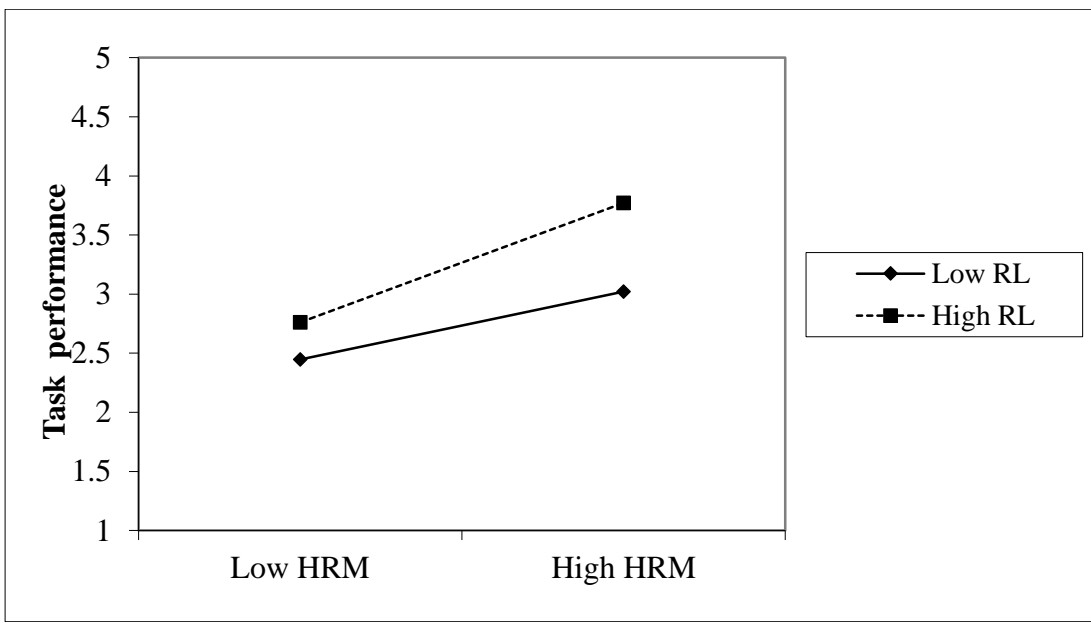

**Figure 4.** The interaction effects of HRM and responsible leadership on employee task performance. HRM: Human resource management; RL: Responsible leadership.

Employee age was significantly and positively associated with task performance (M5: $\beta = 0.11$, $p < 0.05$). Millennial employees include individuals born in the 1980s (born 1981–1990) and 1990s (1991-2000). There is evidence showing that there are differences in work attitudes and behavior between the two groups (Zhu et al., 2015). Thus, the respondents were divided into the two groups (1980s and 1990s). A comparison of the two groups were made using ANOVA tests. The results showed that the 1980s group (n = 50) had significantly higher scores for perceived HRM (F = 61.40; $p < 0.001$), responsible leadership (F = 46.30; $p < 0.001$), employee well-being (F = 65.69; $p < 0.001$), and task performance (F = 46.77; $p < 0.001$) when compared with the 1990s group.

**Table 4.** Hypothesis test results.

| Variables | Employee Well-Being | | | | Task Performance | | | |
|---|---|---|---|---|---|---|---|---|
| | M1 | M2 | M3 | M4 | M5 | M6 | M7 | M8 |
| Age | 0.253 | 0.048 | 0.002 | 0.276 | **0.105** * | 0.074 | 0.035 | 0.035 |
| Gender | 0.056 | 0.050 | 0.046 | −0.093 | −0.130 | −0.102 | −0.105 | −0.129 |
| Tenure | 0.217 | 0.129 | 0.134 | 0.194 | 0.048 | 0.112 | 0.116 | 0.047 |
| Education | −0.026 | 0.018 | 0.050 | −0.033 | −0.016 | 0.024 | 0.050 | 0.025 |
| Position | 0.066 | 0.033 | 0.061 | 0.040 | −0.005 | 0.003 | 0.027 | −0.005 |
| Ownership | −0.018 | −0.026 | −0.019 | 0.008 | 0.020 | 0.006 | 0.011 | 0.021 |
| Size | 0.310 | 0.105 | 0.075 | 0.201 | −0.008 | −0.004 | −0.029 | −0.067 |
| HRM | | 0.294 *** | 0.268 *** | | | 0.391 *** | 0.369 *** | 0.233 *** |
| RL | | 0.397 *** | 0.393 *** | | | 0.270 *** | 0.267 *** | 0.066 |
| HRM*RL | | | 0.130 ** | | | | 0.109 ** | 0.043 |
| Well-being | | | | | 0.676 *** | | | 0.510 *** |
| $R^2$ | 0.34 | 0.59 | 0.61 | 0.31 | 0.61 | 0.53 | 0.54 | 0.64 |
| $\triangle R^2$ | 0.32 | 0.58 | 0.59 | 0.29 | 0.59 | 0.51 | 0.52 | 0.62 |
| F | 17.60 | 38.09 | 35.92 | 14.82 | 44.96 | 29.26 | 27.15 | 37.54 |
| Bootstrap | | | | | | | | (0.39, 0.74) * |

Note. * $p < 0.05$, ** $p < 0.001$, *** $p < 0.001$, two tailed. RL: Responsible leadership; HRM: Human resource management.

## 5. Conclusions and Implications

The main purpose of this research was to explore how HRM and responsible leadership influence millennial employee well-being and task performance in the hospitality industry, revealing how hospitality companies and their management can do more to contribute to millennial employee well-being leading to improved task performance. The results suggested that HRM and responsible leadership have positive effects on employee well-being and, consequently, on task performance. Responsible leadership can strengthen the effects of HRM on employee outcomes. Employee well-being plays a mediating role between HRM and responsible leadership and task performance. Therefore, it is important to improve the millennial employee well-being to facilitate the human resource utilization and hotel sustainable development.

Moreover, this research explored how HRM and responsible leadership affects millennial employees and task performance in hospitality. It demonstrates that HRM and responsible leadership have positive and complementary impacts on employee well-being including life, work, and psychological well-being [25]. This research is a response to the call for "improving health and well-being in society" and to the question of how organizations can contribute more to millennial employee well-being. The analysis focuses on HRM and responsible leadership as critical organizational factors to impact millennial employee well-being in hotels. However, it is not sufficient for HRM and responsible leadership to independently improve employee well-being and performance; more attention must be paid to the interactive effects of HRM and responsible leadership in organizations. Undoubtedly, HRM departments and supervisors need be more responsible in hotels. Further creative and exhaustive research is needed in the future to explore the antecedents of employee well-being involving organizational and individual factors in the hospitality industry.

### 5.1. Theoretical Implications

This research further developed the millennial employee well-being antecedent and outcome research in hospitality organizations. To make people happier in their work is the responsibility of hospitality organizations for improving employee lives. This research provides evidence that millennial employee well-being benefits organizations, and this can be especially important in offering high-quality service in the hospitality industry—it is consistent with the research of Ariza-Montes et al. (2018) [60]. The traditional view of performance is that it has been pursued without adequate concern for employee well-being. However, improvements in employee well-being and higher performance are feasible in a more rapidly changing environment that includes the development of technology and flexible employment, responding to the call of Guest (2017) [27]. Moreover, well-being is increasingly being emphasized by millennial employees [8,17,42]. These research findings provide evidence that employee well-being can facilitate better performance in the workplace and create win–win situations for individual employees and organizations.

This investigation sheds greater light on the interaction effects of HRM and responsible leadership on millennial employee well-being and task performance in workplaces. HRM and leadership overlap in managing human beings within organizations, and the complementary effects is not clear in research (Jiang et al., 2015; Leroy et al., 2018). This study supports the view that HRM and leadership interact in influencing millennial employee well-being in hospitality organizations. HRM and responsible supervisory leadership are antecedents of employee well-being in hotels. HRM and responsible leadership enhanced millennial employee well-being leading to improved task performance. Consistent with job demand and resource perspective and the conservation of resource theory, the results support the view that organizational HRM systems and responsible leadership provide job resources, leading to positive experiences including happiness, self-respect and meaningfulness, and have significant impacts on employee task performance [24,25,40,80]. Contrary to the substitutes-for-leadership model [23], responsible leadership does not reduce the effect of HRM. It is worth mentioning that HRM and responsible leadership are not substitutes for one another but complement each other's influence on millennial employee well-being and task performance. This

research also responded to the call for extending the effects of HRM and leadership on employee well-being in hospitality given the changing workforce [81].

The current analysis extends the research on responsible leadership in hospitality. The results show that responsible leadership should be considered when assessing the contribution of HRM to employee well-being. Responsible leadership is drawing greater attention in research and practice [33,34,39]. However, the effects of responsible leadership on employees remain unclear in the hospitality sector. The available literature highlighted the impacts of responsible leadership on employee satisfaction, creativity, and organizational citizenship behavior (OCB) in the hospitality industry [36–38]. In addition, the accumulated research is based on agency, stakeholder, and institutional theories to explore the effects of responsible leadership [34,35,40,91]. There are limitations to the present leadership with respect to millennial employees in terms of showing concern, being caring and supportive, and building self-respect [21]. In addition, how responsible leadership impacts employee well-being still needs more scholarly attention. This work makes a contribution in its focus on the interaction effects of responsible leadership and HRM on millennial employee well-being, leading to task performance in the hospitality sector, thereby exploring the impacts of responsible leadership on employee well-being and task performance.

### 5.2. Managerial Implications

Hospitality managers should take on the responsibility to improve millennial employee well-being and this is one of the foundations of sustainability in the sector. Millennial employee well-being should be afforded greater attention in hospitality. Positive employee attitudes and behavior stimulate customer satisfaction and loyalty with services, and these are key antecedents of organizational performance. With the increasing millennial involvement in hotel operations, improving millennial employee well-being could be a significant HRM approach for achieving sustainable development, since the millennials will soon become the main workforce. In addition, the 1980s millennials had higher levels of perceived HRM and responsible leadership contributions, well-being, and task performance compared with the 1990s group. It is essential to further strengthen HRM practices and leadership to promote different groups' well-being, especially for the 1990s millennials. Moreover, it is an appropriate strategy to improve millennial employee well-being in HRM in hospitality, where there are high stress levels, work–life conflicts, employee burnout, and high levels of turnover.

HRM should emphasize millennial employee development, growth and commitment relationships between individuals and organizations. Hospitality employees should be viewed as resources not expense items. HRM practices of value-based recruitment and selection, extensive training, performance-based appraisal and compensation, internal promotion and teamwork can help assure employees' competence, development and relationship needs. Employee assistance plans reflect organizational care and concern for staff, and HRM can be a bridge to build harmonious organization–individual relationships.

Line supervisors should show care, support, respect, and ethics in hotels, especially with millennial employees. Responsible leadership provides the day-to day and person-to-person experiences in workplaces besides the formal HRM approaches including training, compensation and performance appraisal. Effective supervisors focus on influencing employees to achieve organizational goals. Caring about and showing respect for staff in decision making are factors that impact employee physical, psychological, and social health. Therefore, the hospitality industry should invest in responsible leadership training for managers.

### 5.3. Limitations and Future Research Directions

It is acknowledged that there are several shortcomings in this analysis. First, the research focus was on employee perceived HRM and responsible leadership and their outcomes. The cross-sectional design is limited in explaining the causality relationship between antecedents and outcomes. In the future,

longitudinal research is needed to explore the causality relationship between HRM and responsible leadership and their outcomes.

The data were collected from employees in hospitality businesses, and using a single source inevitably leads to common variance. An attempt was made to control for common variance bias through the randomizing of items in the questionnaire and by examining whether the common variance bias was acceptable in this research. Future researchers should gather data from multiple sources including managers and employees.

Finally, this investigation did not consider the impacts of other organizational factors. Although leadership and HRM are important antecedents of employee outcomes, other organizational factors should be considered in improving employee well-being in the hospitality sector. In the future, an expanded conceptual model should be designed to test the effects of organizational factors such as organizational culture and dynamic environments.

**Author Contributions:** Conceptualization, J.H. and A.M.M.; methodology, J.H.; software, J.H.; validation, J.H., H.Z. and A.M.M.; formal analysis, A.M.M.; investigation, J.H., H.Z.; resources, H.Z. and A.M.M.; data curation, J.H.; writing—original draft preparation, J.H.; writing—review and editing, A.M.M.; visualization, A.M.M.; supervision, A.M.M.; project administration, J.H. and A.M.M.; funding acquisition, J.H. and H.Z.

**Funding:** This research was funded by Hunan Education department, grant number "18B227", National Social Science Foundation of China, grant number "18CJY009" and Hubei Education department, grant number "15q174".

**Acknowledgments:** We acknowledge the suggestions from Professor Fevzi Okumus.

**Conflicts of Interest:** The authors declare no conflict of interest.

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
