# Peer review of "Improving Millennial Employee Well-Being and Task Performance in the Hospitality Industry: The Interactive Effects of HRM and Responsible Leadership"

_sustainability, doi:10.3390/su11164410_

Round 1
Reviewer 1 Report
The manuscript is to determine the influence of organizations on millennial employee well-being and the task performance in the hospitality industry. The research hypotheses were tested with multiple linear regression analysis based on 21 a survey of 243 millennial employees in the Chinese hospitality industry. The topic is interested. However, there are some suggestions as follows.
1. I suggest that authors to revise the title. “In hospitality” should be “in hospitality industry”.
2. There are a lot of leadership types and different way to classify leadership. The authors should why responsible leadership plays an important role in hospitality industry.
3. Authors mention supervisory leadership is key antecedents of employee well-being. Does supervisory leadership mean responsible leadership in line 60? If supervisory leadership doesn’t mean responsible leadership, please explain the reason why responsible leadership is critical to employee well-being in line 66.
4. Please provide more background information of the respondents, such as respondents’ employers, the locations of respondents’ employers and so on.
5. The authors simply show the reliability of the scale. They need to provide validity data, such as AVE, to show the scale is valid.
6. As mentioned by the authors, the data were collected form employees in hospitality business and a single source inevitably leads to common variance. I suggest that the authors should conduct test if CMV exists in research data.
Reviewer 2 Report
· Summary of the manuscript
The purpose of this manuscript is to investigate the interactive effects of human resource management (HRM) and responsible leadership on millennial employees’ (born between 1981 and 2000) well-being and performance particularly in hospitality industry. The reason why the well-being of millennial employees is important in hospitality industry is that they are becoming main forces in that industry. Based on the survey of 243 millennial employees in the Chinese hospitality industry, the authors found that HRM and responsible leadership show positive associations with both millennial employees and their performances. Although the manuscript was well-written in general, the review believes that the research must take another way of design. It seems too critical to convince the robustness of the finding and implications of the research.
· General comments
- First, the reviewer thinks that the hypothesized model (Fig. 1) should be located in or after Section 2 including the hypotheses driven in Literature Review and Hypotheses (Section 2).
- Second, the reviewer believes that in order to examine the interactive and mediating effects of millennial employees’ well-being, the well-known structural equation modelling should be more adequate than multiple regression in current form. Otherwise, the two-stage least square regression must be conducted to prevent endogenous issues between the variables.
- Third, the findings are believed to give useful implications to both managers and researchers. However, the methodological issues attenuate the implications of them.
Round 2
Reviewer 1 Report
The authors haved made nccessary change that suggested by reviewer, I think the manuscript could be accepted.
Author Response
We note that Reviewer 1 has already indicated "Accept."
Reviewer 2 Report
As a reviewer, I really appreciates the authors' efforts to reflect my comments.
However, my concern about the research methodology does NOT disappear after the authors’ response. The papers cited for justifying the methodology, which are “Liu, Zhang & Wang (2012)” and “Kalshoven and Boon (2011)”, CANNOT justify the methodological issue.
First, the paper “Liu, Zhang & Wang (2012)” does NOT appear in the list of references. The review is able to verify whether the authors adopted a proper methodology or not. Please update the list of reference and let me know what the referred paper deals with.
Second, the other paper “Kalshoven and Boon (2011)” particularly focuses on multilevel analysis where the data were collected in individual level but used in individual and/or organizational levels. However, this manuscript is NOT a multilevel analysis. In this manuscript, the data were collected in individual level and there are no ICC indicators.
Following the two comments above, I have a bigger concern about the methodological issue of this manuscript.
Author Response
appended are our responses to Reviewer 2's comments.

Round 3
Reviewer 2 Report
As a reviewer, I again appreciates the authors' explanation.
Although the authors explained that the methodology used in this research has been accepted by various researcher (50 citations counted in Google Scholar), I CANNOT agree with the scientific soundness of the research.
I believe that the motivation, results, and implications are important and interesting in both practical and theoretical contexts. However, I believe that structural equation modelling is more appropriate to investigate the moderated mediation effect of the theoretical model suggested in this manuscript.
I recommend the authors to adopt other sound methodology rather than the controversial one used in current form.